# Mistreatment of Women during Childbirth and Associated Factors in Northern West Bank, Palestine

**DOI:** 10.3390/ijerph192013180

**Published:** 2022-10-13

**Authors:** Ibtesam Medhat Mohamad Dwekat, Tengku Alina Tengku Ismail, Mohd Ismail Ibrahim, Farid Ghrayeb, Eatimad Abbas

**Affiliations:** 1Department of Community Medicine, Universiti Sains Malaysia, Kota Bharu 16150, Kelantan, Malaysia; 2Faculty of Health Professions, Al-Quds University, Jerusalem 51000, Palestine

**Keywords:** mistreatment during childbirth, obstetric violence, women’s rights

## Abstract

Mistreatment of women during childbirth is a clear breach of women’s rights during childbirth. This study aimed to determine the prevalence and associated factors of mistreatment of women during childbirth in the north of West Bank, Palestine. A cross-sectional study was conducted among 269 women within the first 16 weeks of their last vaginal childbirth to understand the childbirth events by using proportionate stratified random sampling. An Arabic valid questionnaire was used as a study instrument. Simple and multiple logistic regression analyses were conducted to determine the factors associated with each type of mistreatment. The mean age of the women was 26.5 (SD 4.77) years. The overall prevalence of mistreatment was 97.8%. There were six types of mistreatment. Nine factors were significantly associated with the occurrence of one or more types of mistreatment. Delivery at a public childbirth facility was associated with all of the six types (aAdjOR: 2.17–16.77; *p*-values < 0.001–0.013). Women who lived in villages (aAdjOR 2.33; *p*-value = 0.047), had low education (aAdjOR 5.09; *p*-value = 0.004), underwent induction of labour (aAdjOR 3.03; *p*-value = 0.001), had a long duration of labour (aAdjOR 1.10; *p*-value = 0.011), did not receive pain killers (aAdjOR: 2.18–3.63; *p*-values = 0.010–0.020), or had an episiotomy or tear (aAdjOR 5.98; *p*-value < 0.001) were more likely to experience one or more types of mistreatment. With every one-hour increase in the duration of labor, women were 1.099 times more likely to experience a failure to meet the professional standard of care. Women were less likely to experience mistreatment with increasing age. Women with increasing age (aAdjOR: 0.91–0.92; *p*-values = 0.003–0.014) and parity (aAdjOR 0.72; *p*-value = 0.010) were less likely to experience mistreatment. Awareness of women’s fundamental rights during childbirth, making the childbirth process as normal as possible, and improving the childbirth facilities’ conditions, policies, practices and working environment may decrease mistreatment occurrence.

## 1. Introduction

In spite of the extent of mistreatment of women during childbirth which contributes to a significant breach of women’s fundamental rights [1,2], mistreatment is still insufficiently addressed under international human rights law [2]. In seeking and receiving care before, during and after childbirth, every woman is entitled to several rights. They include the rights to: (1) be free from harm and ill-treatment; (2) information, informed consent and refusal, and respect for a woman’s choices and preferences, including companionship during childbirth; (3) privacy and confidentiality; (4) treatment with respect and dignity; (5) equality, non-discrimination, and equitable care; (6) healthcare and high achievable level of health; and (7) liberty, autonomy, self-determination and non-coercion [3].

Unfortunately, the presence of mistreatment of women during childbirth has been confirmed globally [4,5,6,7,8,9,10,11,12,13,14]. Evidence showed that a third of women are exposed to verbal or physical abuse during childbirth [4].

Mistreatment of women during childbirth is a complicated phenomenon and its definitions are heterogeneous across the literature due to the variation in the types and terms used. The most commonly employed term is “disrespect and abuse”, which was introduced in landscape analysis [5]. It was defined as “interactions or facility conditions that local consensus deem to be humiliating or undignified, and those interactions or conditions that are experienced as or intended to be humiliating or undignified” [6]. 

The term “mistreatment” can be used instead of “disrespect and abuse” as it is more comprehensive because it covers a wide scope of categories and emphasizes different sources of mistreatment [7]. “Obstetric violence” is another term that is also used in Latin America and the Caribbean to describe abuse or mistreatment during childbirth that women face from healthcare providers [8]. The three terms; “obstetric violence”, “disrespect and abuse” and mistreatment are shared concepts (medicalization of the childbirth process, gender inequality) alongside violence against women [9]. These mentioned three concepts are used to explain the extent of mistreatment.

The use of seven evidence-based types of mistreatment was recommended in a systematic review that encompassed physical abuse, sexual abuse, verbal abuse, failure to meet professional standards of care, stigma and discrimination, poor rapport between women and providers, and healthcare system conditions and constraints [7]. These types are recommended to be considered when developing new tools for better measuring and for avoiding underestimation of the prevalence of mistreatment [7]. 

Global mistreatment prevalence ranged from 11 to 98%, as reported in previous studies (e.g., 98% in Nigeria [10], 78% in Kenya [11], 97% in Pakistan [12], 28.8% in India [13], 49.4% in Latin America [14], 18.3% in Brazil [15], and 11% in Mexico [16]). Mistreatment prevalence may vary due to cultural differences in the participants’ setting of origin, in addition to childbirth conditions and restraints. Moreover, the manifestations and detailed assessment of the types of mistreatment presented, as well as the timing and place of data collection, influenced the reported prevalence [17]. 

Certain factors were found to be associated with the mistreatment of women during childbirth. These factors included being a young-age mother [4,15,18,19] or being a single mother [20], being uneducated [4,10], having low socioeconomic status [10,21,22] and living in poor residency [23]. On the other hand, obstetric characteristics, such as the presence of complications during childbirth, sex of the provider [13] and women’s parity were also reported as factors associated with the occurrence of mistreatment [7,11,13]. An additional factor was giving birth at a public childbirth facility [7,13,15,24].

### Palestinian Context

Palestine is a country which is located in Asia between the Mediterranean Sea and the Jordan River. Historically, it is known that a part of Palestinian land has been occupied by Israel since 1948. Since 1948, the political situation in Palestine has been unstable and the Palestinian people have been striving to obtain their fundamental rights as human beings and to take back their land and live in peace and dignity. 

In 1967, Israel occupied the rest of Palestine which includes West Bank and Gaza which are now known as the Occupied Palestinian Territory (OPT). The Palestinian Authority (PA) was founded in 1994 and which was given some control over the OPT land and was responsible for governing the health care system. The United Nations Relief and Works Agency (UNRWA) was founded by the United Nations in 1949 as a consequence of Israeli occupation for the purpose of providing direct relief and works programs for Palestinian refugees. Since that time, the UNRWA has shared in the Palestinian health care system. 

The political situation in Palestine has negatively affected the healthcare system, especially the public childbirth facilities. Therefore, this system became very poor with low quality [25], chronic deficiency of medical supplies and vital medical disposable, which affects certain areas and treatment pathways [26]. 

In fact, public childbirth facilities are considered the convenient choice for women with low- and middle-income because of the availability of health insurance coverage while women with high income utilize private childbirth facilities. The financial status of the family plays an important role in a woman’s choice of facility. 

Unfortunately, the Palestinian community has special considerations due to the political situation, thus Palestinians always suffer from a violation of human rights and experience feelings of fear, worry, and threats in their daily life [27]. With regards to the Palestinian women who often suffer from this Israeli occupation, they have extra difficulty in their lives because when experiencing labor pain, women usually have an unsafe and insecure way to reach childbirth facilities due to the presence of roadblocks [25]. Previous literature also showed that in the period from 2000 to 2006, 69 Palestinian women delivered their babies at checkpoints in West Bank under unsafe and dehumanizing conditions [28].

Israeli occupation and the political instability in Palestine are one of the main hiding factors of the abusive mistreatment of Palestinian women during their lives, especially during childbirth. Mistreatment during childbirth is a part of the violence against women that women may be exposed to in their lives and is also considered a major violation of women’s fundamental rights. Actually, Palestinian women have unique experiences, facing multiple types of abusive mistreatment at the same time. Investigating mistreatment among these women is very necessary as it will be the first step in raising awareness of the fundamental rights that women are deprived of due to political conflict and help towards ending serious and abusive mistreatment of Palestinian women at a vulnerable time in their lives. The aim of this study is to measure the prevalence of the mistreatment of women during childbirth and its associated factors in the northern West Bank, Palestine.

## 2. Materials and Methods

### 2.1. Study Design, Setting and Sample

A cross-sectional study was conducted between October 2020 and January 2021. The main maternal and child health clinics belonging to the Palestinian Ministry of Health in the northern area of the West Bank were selected as the study locations. These clinics are located in the six governorates of Jenin, Nablus, Tulkarm, Tubas, Qalqelia, and Salfeet. The inclusion criteria included women during the first 16 weeks of their last vaginal childbirth, aged 18 years or older, and who visited the abovementioned clinics. Those women with complications after childbirth, multiple pregnancies, stillbirth or neonatal death during their last childbirth, or known psychiatric illness were excluded from the study. 

The sample size was calculated using a single proportion formula considering a prevalence of mistreatment during childbirth of 20% [24], a confidence interval of 95%, and an absolute degree of precision of 5%. Consequently, the estimated sample size was 245. After allowing for a non-response rate of 10%, the total sample size was set at 269 participants. Proportionate stratified random sampling was used to obtain the required number of participants from each governorate. The calculation was based on the total number of deliveries per governorate, considering that Palestinian women usually give birth at several facilities within the same governorate in which they live. The aim of using proportionate stratified random sampling was to acquire a sample that was representative of the whole population and to ensure that women in each governorate were adequately represented and had an equal chance of being selected for the study.

### 2.2. Study Instrument

A pretested self-administered mistreatment during childbirth questionnaire, developed in the Arabic language and validated among women during their first 16 weeks postpartum in West Bank, Palestine, was developed through the qualitative study and used for the purpose of measuring the experience of mistreatment of women during childbirth, the mistreatment types, and its associated factors [29]. The questionnaire development was based on a literature review and findings from a qualitative study, while the validity of the questionnaire was confirmed by considering content validity, face validity, and factor analysis [29,30]. The domains of the questionnaire utilized to fulfill the study objectives were socio-demographic, obstetric, and childbirth history (18 items), as well as the woman’s experience of mistreatment during childbirth (43 items). The experience of mistreatment during childbirth domain was represented by six types of mistreatment: (1) physical abuse; (2) verbal abuse; (3) stigma and discrimination; (4) poor rapport between women and providers; (5) failure to meet professional standards of care; and (6) health system conditions and constraints. Sexual abuse was not included in this domain due to the sensitivity of the subject among the Palestinian community. Furthermore, this type of mistreatment was not expressed by the respondents of the qualitative study during the development of the questionnaire [29,30]. For more detailes about the questionnaire items and domains, please see Appendix A.

Out of the six types of mistreatment, three of them (poor rapport between women and providers, failure to meet professional standards of care, and health system conditions and constraints) comprised several subtypes, as demonstrated in Table 1. For example, statements pertaining to the poor rapport between women and providers expressed the lack of communication between women and healthcare providers, lack of supportive care (e.g., the presence of a birth companion, encouragement, and reassurance), and loss of autonomy (e.g., statements regarding the women’s participation in decision-making and care provided to them and the way healthcare providers handled women’s bodies). There were negative and positive statements in this questionnaire. Examples of the types that included negative statements were lack of informed consent and explanation during childbirth and violation of confidentiality. The mistreatment type that included both negative and positive statements was the negligence of care. With regard to the conditions and constraints of the healthcare system, all the statements were positive. 

Options of “yes”, “no”, or “not applicable” were used for each statement in the respective mistreatment types. Those who answered “yes” to any of the items regarding each type were considered as having experienced the respective type(s) of mistreatment during childbirth, taking into consideration the reversed statements. 

### 2.3. Data Collection and Procedures

Data were collected for a period of three months from all six governorates in the northern area of the West Bank beginning 1 October 2020 until 31 January 2021. Data collection was conducted concurrently using three midwives as data collectors under the supervision of the main researcher. The number of questionnaires distributed in each governorate was based on the calculated ratio previously explained with a response rate of 100%. The first data collector recruited 75 participants from the governates of Jenin and 13 from Tubas. The second data collector recruited 91 participants from Nablus, and the third data collector recruited 42 participants from Tulkarm, 19 from Salfeet, and 29 from Qalqelia. The data collectors met the targeted women at the main maternal and child health clinics belonging to the Palestinian Ministry of Health in each governorate. The Palestinian Ministry of Health clinics are considered the main centers for children’s immunization and these centers can be utilized by women coming from both public and private childbirth facilities. 

Before the study was conducted, the data collectors were trained in the correct method of collecting data, how to understand the questionnaire, and in the proper way of interacting with the participants. The data collectors were also provided with instructions about the objectives of the research, the gathering of sensitive data, and applying ethical principles during data collection. This training was conducted to ensure the standardization of the data collection procedures. The women were approached by the data collectors at the maternal and child health clinics during their visits to vaccinate their newborns. Actually, the interactions between the midwives and the women may include some bias and subjectivity as well as recruiting the women within the Palestinian Ministry of Health clinics may also affect the representativeness of the sample. The participants were given explanations about the study objectives and were reassured that participating in this study was voluntary and that non-participation would not affect their receiving services. Written consent was obtained once they agreed to participate. The questionnaire was distributed to the targeted women in paper format. They completed the questionnaire and the copies were collected on the same day. The average time to complete the questionnaire was 20 to 25 min. The process continued until the required number of participants for each clinic was obtained. Ethical approval for this study was obtained from the Human Research Ethics Committee of Universiti Sains Malaysia (reference number: USM/JEPeM/18080400) on 29 November 2018; this study was also accepted by the ethical committee at Al-Quds University at West Bank, Palestine (reference number: 57/REC/2018) on 27 November 2018. Additionally, permission from the Palestinian Ministry of Health Administration in the West Bank was obtained for this study, including for the sample, methodology, and data collection site.

### 2.4. Statistical Analysis

All 269 copies of the questionnaire were entered into the IBM Statistical Program for Social Sciences (SPSS), version 25, by the main researcher. The data entry was randomly checked by the other research team member. The data were stored in password-protected computer files, and hard copies of questionnaires were secured. All the data were checked and cleaned initially; data screening was conducted for missing values or possibly incorrect data entry before analysis. The descriptive findings were presented in terms of frequency, percentage, mean, and standard deviation. The frequencies and percentages of each type of mistreatment were calculated after reversing all the positive statements. 

With regard to the associated factors of mistreatment, simple and multiple logistic regression analyses were conducted. Simple logistic regression analysis was utilized first to identify the important independent variables to be employed in the multiple logistic regression analysis. Variables with a *p*-value < 0.25 and those of clinical importance were included. The clinically important variables were determined by experience, clinical data, and previous studies. Odds ratios were determined at a 95% confidence interval (CI). The model fitness was tested using the Hosmer–Lemeshow goodness-of-fit test [31]. The model is considered fit if the *p*-value is greater than 0.05. Interactions between the variables were checked for every model as well as any possible two-way interactions were also investigated. Multicollinearity was examined by using correlation matrix and standard error. The associations of socio-demographic, obstetric, and childbirth characteristics with the six types of mistreatments were analyzed independently. The independent variables were age, education, household income, residency, occupation, parity, nature of labor, duration of labor, receiving pain killer, type of delivery, time of delivery, sex of providers who conducted the delivery, and types of facility. Each type of mistreatment was the dependent variable; therefore, six simple and multiple logistic regression analyses were conducted.

## 3. Results

### 3.1. Socio-Demographic Characteristics of the Participants

A total of 269 married Palestinian women participated in the study. They were aged from 18 to 41 years old, with a mean age of 26.5 (SD 4.768) years. A total of 115 (42.7%) had received either secondary education or less, while 154 (57.3%) had attained higher education. With respect to residence, 135 (50.2%) came from cities, 129 (48.0%) from villages and five (1.8%) lived in camps. Their mean monthly household income was 3172 NIS (SD 1582) (Table 2).

### 3.2. Obstetric and Childbirth Characteristics

The participants’ parity ranged from one to eight. The highest frequency was a parity of two to three, 124 (46.1%), followed by a parity of one, 87 (32.3%) and then a parity of four or higher, 57 (21.3%). Around 110 (40.9%) were in the first seven weeks postpartum, and 159 (59.1%) were 8–16 weeks postpartum. All the participants had their current childbirths at childbirth facilities, with a majority of 169 (62.8%) at public childbirth facilities. The majority of the deliveries were conducted by female healthcare providers at 215 (79.9%). Only 129 (48%) of the participants received painkillers during childbirth. More than half underwent spontaneous labor (56.1%) and 149 (55.4%) delivered through vaginal delivery with no episiotomy or tear (Table 3).

### 3.3. Prevalence and Types of Mistreatment Women Experienced during Childbirth

The overall prevalence of mistreatment was 97.8% (95% CI: 95.9, 99.5). Among the six types of mistreatment, the poor rapport between women and providers was the most commonly reported (88.8%). This included ineffective communication, lack of support, and loss of autonomy during childbirth. The second most common type was physical abuse (76.6%), which frequently manifested as painful vaginal examinations and the application of abdominal pressure by the providers during delivery. Failure to meet a professional standard of care was the third most commonly reported type (75.8%); this included the lack of informed consent and explanation about various procedures during childbirth, violation of confidentiality, and negligence of care. The other types of mistreatment were verbal abuse (24.5%), health system conditions and constraints, including a lack of physical privacy during childbirth (22.3%), and stigma and discrimination (11.9%) (Table 4).

### 3.4. Factors Associated with Mistreatment during Childbirth

Table 5 represents the simple logistic regression analysis for factors associated with the six types of mistreatment that women encountered during childbirth. From the analysis, 13 independent variables were found to be associated with at least one type of mistreatment, with *p*-values < 0.25. These variables were age, education, monthly income, residency, occupation, parity, nature of labor (spontaneous/induction), duration of labor (hours), receiving painkillers, type of delivery (vaginal with no episiotomy or tear, vaginal with episiotomy or tear), time of delivery (day/night), sex of providers conducting the delivery, and type of facility (public/private). Moreover, the mentioned variables were also clinically significant and were demonstrated in the literature to play important roles in women’s path of care during childbirth; thus, they were selected for multiple logistic regression analysis. Multiple logistic regression analysis resulted in nine significant variables which were age, education, residency, parity, type of facility, nature of labor, type of delivery, duration of labor, and receiving painkillers (Table 6). For more detailes about the multiple logistic regression analysis, please see Appendix A.

Age, nature of labor and type of facility were found to be significantly associated with physical abuse. For each one-year increase in age, the women were 8.9% less likely to experience physical abuse (95% CI: 0.86, 0.97, *p*-value = 0.003). Those who were subjected to induction of labor had 3.03 times higher odds of physical abuse compared to women with spontaneous labor (95% CI: 1.58, 5.81, *p*-value = 0.001). Additionally, women who delivered at public childbirth facilities were 2.17 times more likely to experience physical abuse compared to those whose delivery took place at private childbirth facilities (95% CI: 1.18, 4.02, *p*-value = 0.013).

Age and type of facility were also significantly associated with verbal abuse. For each one-year increase in age, the women were 7.8% less likely to experience verbal abuse (95% CI: 0.86, 0.98, *p*-value = 0.014). Those who delivered at public childbirth facilities had 3.25 times the odds of experiencing verbal abuse compared to women who gave birth in private facilities (95% CI: 1.65, 6.41, *p*-value = 0.001).

Type of facility, type of delivery and residence were the significant factors for stigma and discrimination. Delivery at public childbirth facilities put the women at a 16.78 times higher likelihood of being discriminated against during childbirth compared to those who delivered at private facilities (95% CI: 3.79, 74.26, *p*-value < 0.001). In addition, women who had an episiotomy or tear during vaginal delivery were 5.98 times more likely to be discriminated against than those who did not have one (95% CI: 2.52, 14.15, *p*-value < 0.000). Women from villages had 2.33 times the odds of being discriminated against during childbirth compared to those living in cities (95% CI: 1.01, 5.37, *p*-value = 0.047).

The type of facility and duration of labor were also found to be associated with a failure to meet professional standards of care. This indicated that women who delivered at public childbirth facilities had a 3.31 times higher chance of experiencing a lack of consented care, violation of confidentiality, or negligence of care than those who delivered at private childbirth facilities (95% CI: 1.83, 5.97, *p*-value < 0.001). Additionally, with each one-hour increase in the duration of labor, women were 1.099 times more likely to experience a lack of consented care, violation of confidentiality, or negligence of care (95% CI: 1.02, 1.18, *p*-value = 0.011).

Type of facility, education, or receiving painkillers was significantly associated with the poor rapport between women and providers. Delivery at public childbirth facilities put the women at 4.61 times higher risk of encountering ineffective communication, lack of supportive care, or loss of autonomy than delivery at private childbirth facilities (95% CI: 1.82, 11.71, *p*-value = 0.001). Those with secondary education or lower were 5.09 times more likely to face ineffective communication, a lack of supportive care, or loss of autonomy than women with higher education (95% CI: 1.67, 15.57, *p*-value = 0.004). Additionally, women who did not receive painkillers during childbirth had 3.63 times higher odds of experiencing ineffective communication, a lack of supportive care, or loss of autonomy than women who received them (95% CI: 1.36, 9.71, *p*-value = 0.010).

Likewise, the type of facility, receiving painkillers, and parity were significantly associated with a lack of physical privacy and resources. Women who delivered at public childbirth facilities had 2.75 times higher odds of facing a lack of physical privacy and resources than women who delivered at private childbirth facilities (95% CI: 1.33, 5.68, *p*-value = 0.007). Those who did not receive painkillers during childbirth were 2.18 times more likely to experience a lack of physical privacy and resources during childbirth compared to women who received painkillers (95% CI: 1.13, 4.22, *p*-value = 0.020). With each unit increase in parity, women were 27.6% less likely to experience a lack of physical privacy and resources (95% CI: 0.57, 0.93, *p*-value = 0.010).

## 4. Discussion

A total of 269 women in their first 16 weeks post-vaginal birth were included in the study; the women were randomly chosen from the six governorates located in the northern area of the West Bank using proportionate stratified random sampling. This sampling method allowed for the ratio of women post-delivery in each governorate to be preserved. This was important to ensure that the sample represented the entire population since the number of women delivering differed by governorate.

The majority of the women in this study (97.8%) experienced at least one type of mistreatment during childbirth. Similar findings were reported by studies in Asia (97%) [12] and Africa (98%) [10]. The presented high level of mistreatment constitutes a major violation of women’s rights [2,6]. Actually, highlighting mistreatment and raising awareness of women’s rights during childbirth is a very important step to help women exercise their rights. 

Multiple hidden factors play an important role in the extent of the mistreatment of women in Palestine and the violation of women’s rights during childbirth. These factors include the political situation in Palestine, the structure of the Palestinian health care system and its futile policies, medicalization to normal childbirth process and society culture, beliefs, gender inequalities and male dominance.

The political situation in Palestine and its negative effects on the health care system resulted in poor public childbirth facility environments. The problems related to poor childbirth facility environments are common and reported not only in the Palestinian healthcare system but also in other countries, especially developing ones [5,7,10,13,21,24,32,33].

In this study, delivery at a public childbirth facility was found to be associated with all six types of mistreatment; physical abuse, verbal abuse, stigma and discrimination, poor rapport between women and providers, failure to meet professional standards of care, and health system conditions and constraints. This means several of a woman’s rights are violated when she gives birth at a public childbirth facility as she is exposed to the six types of mistreatment. Regardless of the numerous health services provided by public childbirth facilities, they are still facing long-standing insufficiency of medical supplies and vital medical disposables, which interferes with certain areas of care and treatment pathways [26]. Moreover, the public childbirth facilities are still used by considerable number of of Palestinian women because of low cost and availability of health insurance coverage eventhough these facilities were not the prefered choice for them [34]. In fact, there is no isolation between the poor conditions of the health care system in the West Bank from the political situation in Palestine because the Israeli occupation is controlling the resources and putting restrictions on the improvement and outside funding. The public childbirth facilities also lack good quality services due to the high number of deliveries and crowded labor rooms [28,35]. This may result in overloaded work for the healthcare providers, which affects their performance and is reflected in the quality of care provided to women. Additionally, a recent policy prevents the presence of a birth companion during delivery; thus, the women have to spend all the time alone in the labor room and feel neglected. Previous studies support the present study’s finding that public childbirth facilities were associated with a high prevalence of mistreatment during childbirth [7,13,15,24].

Furthermore, the structure of the Palestinian health care system and its futile policies are other contributing factors to the mistreatment of women during childbirth. For instance, a policy that prevents the presence of a birth companion exists in public childbirth facilities which women have complained about for a long time. Denial of companionship during childbirth was shown to cause women to feel disempowered and lonely [7]. African women described the disallowing of a childbirth companion as a “crime against humanity” [10]. In addition to that, the presence of ineffective monitoring systems, the lack of accountability mechanisms, and the non-adherence of healthcare providers to evidence-based practices during the provision of care may worsen the situation [5,7,12,14,29].

Giving pain relief to a woman is a very important action to help decrease her suffering during childbirth and it is also one of her fundamental rights in case of the presence of pain. In this study, more than half of the participants did not receive painkillers, which extends the mistreatment of women during childbirth. In fact, the deprivation of painkillers from women during childbirth is one form of mistreatment that is linked to failure to meet professional standards of care [7]. Other previous studies linked not giving painkillers to women during childbirth to poor communication by healthcare providers as they react poorly to labor pain and underestimate the women’s sensation of pain. Thus, the women considered healthcare providers as uncooperative and unfriendly. These results are consistent with the findings of studies conducted elsewhere [33,36,37]. Women who did not receive painkillers when needed would experience more pain and difficulty in childbirth, reflecting a negative childbirth experience. A previous qualitative study conducted among women showed how giving pain relief is an important, necessary element as it decreases their pain and stress during childbirth [38]. Adopting and focusing on the policy of reducing pain relief during childbirth at Palestinian public childbirth facilities is one of the important steps that decrease women’s burdens during this sensitive period as well as pain relief being one of their rights.

Medicalization of normal childbirth processes highly contributes to the extent of mistreatment as it stands side by side with the concept of obstetric violence because it makes women more vulnerable to painful, harmful and difficult procedures [8], such as induction of labor and episiotomies, frequent vaginal examination [32,39] and long labor duration [39,40]. Medicalization of the childbirth process involves denying women’s right to autonomy and consent [2,6]. This also corresponds with the result of this study, in that the nature of labor, especially induced labor, is significantly associated with physical abuse. This is because women who undergo induction of labor are frequently forced to undergo various difficult interventions and procedures that have the nature of mistreatment. These situations were also mentioned in previous related studies in Jordan [33,36].

The longer the duration of labor, the more likely women are to experience negligence of care during childbirth, lack of confidentiality, and non-consented care. This also agreed with the findings of a Swedish study, showing that prolonged labor is a contributing factor to negative childbirth experiences [40].

Furthermore, having a vaginal delivery with the experience of an episiotomy or a tear is also another associated factor of mistreatment. This may be explained due to women losing control over their bodies and also feeling that their rights were disregarded such as privacy and autonomy during an episiotomy procedure, adding to the pain women experience during and after this procedure. Similar complaints were also reported by a recent Chinese qualitative study. Women who experienced an episiotomy reported that they were being criticized by healthcare providers for not tolerating the pain during the procedure [37]. Other studies discovered that women who experienced perineal trauma were neglected by healthcare providers during the procedure [41,42]. In fact, healthcare providers sometimes use the power they receive from the healthcare system when dealing with women during childbirth to control them. Actually, healthcare providers taking coercive procedures on women clearly shows an abuse of power by caregivers, which is a much more serious violation of women’s autonomy and rights [2,6].

The extent of mistreatment of women during childbirth is also affected by society’s culture, beliefs, gender inequality and male dominance, which is very high among the Palestinian community [43]. Actually, the predominant patriarchal culture in Palestinian society is the basis for gender inequality, which limits women from practicing their rights during childbirth as decisions are mainly decided on their behalf. In this study, the age of women, education and parity were found as protective factors against mistreatment. Older women were less likely to experience physical and verbal abuse. The fact that normalization of verbal and physical abuse among older women makes them more adaptive and less sensitive to tough events during childbirth [5,7]. Some women even considered abuse as a way of accelerating their delivery [5,10,11]. In spite of excluding women younger than 18 years from this study, the results nonetheless showed that the younger the woman, the more exposed she is to mistreatment. Actually, women with younger ages, low education and lack of childbirth experience are more prone to mistreatment because of the inside concept from the healthcare providers that these women could be easily controlled referring to their characteristics mentioned above.

Additionally, with an increase in parity, the women were more habituated to the childbirth process and frequent procedures; thus, they had become less sensitized to the lack of privacy and resources. Their expectations of the care that they would receive during childbirth were lowered. The women who were experienced in childbirth had become accustomed to mistreatment and consider it normal [5]. Moreover, these women felt confident, had positive attitudes, and enjoyed higher satisfaction because they had the chance to participate in the decision-making for their care [44,45]. Actually, the root of the normalization of abuse among women during childbirth is derived from society’s culture and gender inequality, which make women accustomed to being abused during their lifetime. 

Education is another factor associated with mistreatment, as reported by previous studies [7,23,46]. Women with a low level of education are associated with poor rapport with providers because the providers thought that women might not comprehend the instructions given to them. Therefore, the healthcare provider intentionally displayed abusive behavior to control the women during childbirth [46]. Accordingly, the invisible causes of mistreatment that women are exposed to during childbirth because of their young ages, their low education and their lack of childbirth experiences, are due to the predominant socialization of men and women into naturalized, forms of violence and power dynamics between groups [8]. This form of mistreatment is also parallel to violence against women, therefore the healthcare providers who hold the power in the labor room unintentionally abuse women through their authority and at other times when healthcare providers decide on behalf of the women during the childbirth process.

The residential area was also a predictor of mistreatment [23]. Residency of the Palestinian women is naturally a mistreating factor for them because of the Israeli occupation and political instability; the presence of roadblocks and checkpoints that women have to pass so that they can reach the childbirth facilities are significant obstacles for them. Previously, some childbirth has taken place at checkpoints which are dehumanized and unsafe environments [25,28]. Such practices are incongruent with human rights, as women must have the right to access safe and respectful care. Women from rural areas are also affected by a lack of access to good medical resources which is a result of chronic shortages in the Palestinian healthcare system [26] because the healthcare system is related—in one way or another—to the political situation in Palestine. 

One of the strengths of this study is addressing the topic itself and its importance for increasing awareness of Palestinian women’s rights during childbirth. It was conducted using a valid and reliable questionnaire that was specifically prepared to achieve the purpose of the study. Furthermore, the data collection was not carried out in the facilities where the women had undergone childbirth; rather, a comfortable environment was provided for them to express their feelings. Moreover, the women chosen to participate in the study were within 16 weeks post-childbirth because they were expected to have recovered from the pain of childbirth and, thus, to be more alert and logical in their responses.

With regards to the limitations of the study, women who were under United Nations Relief and Works Agency (UNRWA) clinics were not included because of the COVID-19 pandemic and the difficulty of obtaining permission from the administrative unit. Findings from these women might differ from those obtained from other women; thus, mistreatment among them should be investigated in the future. In addition, this study only included women who experienced vaginal delivery without complications. Different factors might have been found among those who delivered through cesarean sections or vaginal deliveries with complications. Furthermore, the inclusion of only adults aged 18 years and older due to consent-related matters may result in an inability to understand relevant factors among adolescents. Thus, future studies should be conducted among these groups of women to obtain a clear picture of mistreatment and its associated factors. In addition, there might be differences across the specific childbirth facilities that were not addressed in this study, thus considering this factor in future analysis and studies would be of great importance.

## 5. Conclusions

The results of this study, which is one of the first to consider such issues in West Bank, showed that a high percentage of women experienced mistreatment during childbirth. The most common type was the poor rapport between women and providers, followed by physical abuse and failure to meet a professional standard of care. Age, nature of labor, type of facility, type of delivery, residency, duration of labor, education, receiving painkillers and parity were the factors significantly associated with mistreatment during childbirth. The mistreatment of women during childbirth is caused by multiple factors that negatively affect women’s childbirth experiences. To address the mistreatment revealed by this research, multiple initiatives should be undertaken, such as consideration of the results by stakeholders in the improvement of the environment surrounding childbirth, as well as a massive investment of effort in addressing the factors that lead to the environment of mistreatment, such as staff constraints and poor working conditions. Decision makers should intensify their focus on upgrading childbirth facilities, especially public ones, by improving the childbirth environment in general and the healthcare providers’ working conditions specifically. Related measures may include increasing the number of healthcare providers and motivating them to be more productive and have a more positive attitude toward their work. Additionally, it is vital to improve the conditions surrounding childbirth practices by decreasing unnecessary interventions and striving for a more spontaneous childbirth process. Healthcare providers should provide adequate analgesia during and after the episiotomy procedure to reduce women’s pain and suffering. They must not only acknowledge the delivery of women’s rights but also advocate for those rights.

In addition, policies should be modified in favor of considering women’s preferences during childbirth, such as allowing the presence of a companion and keeping childbirth practices free of unnecessary interventions and making the process of childbirth as normal as possible. Administrators and providers should stress certain vital principles during childbirth, such as respectful care, pain management, communication skills, ethical principles, and women’s rights. A greater concentration is needed on the establishment of systems of monitoring and ensuring effective accountability at childbirth facilities.

## Figures and Tables

**Table 1 ijerph-19-13180-t001:** The types, subtypes, and positive and negative items included in mistreatment of women during the childbirth experience.

	Types of Mistreatment	Subtypes of Mistreatment	No. of Positive Items	No. of Negative Items	Total Items
Mistreatment of Women during Childbirth Experience (43 Items)	Physical abuse	-	-	5	5
	Verbal abuse	-	-	4	4
	Stigma and discrimination	Discrimination due to age and parity	-	6	6
	Failure to meet professional standards of care	Lack of informed consent and explanation	2	2	4
		Violation of confidentiality	-	4	4
		Negligence of care	2	3	5
	Poor Rapport between women and providers	Ineffective communication	2	2	4
		Lack of supportive care	2	1	3
		Loss of autonomy	2	1	3
	Health system conditions and constraints	Physical privacy during examination and during delivery	5	-	5

**Table 2 ijerph-19-13180-t002:** Socio-demographic characteristics of the participants (n = 269).

Variables	Mean SD	n (%)
Age	26.5 (4.768)	
Marital status Married Single Residency		269 (100.0) 0 (0)
City		135 (50.2)
Village		129 (48.0)
Camp		5 (1.8)
Educational		
Secondary or less		115 (42.7)
Higher than secondary school		154 (57.3)
Occupation		
Working		33 (12.3)
No formal work		235 (87.7)
Monthly household income	3172 (1582)	

**Table 3 ijerph-19-13180-t003:** Obstetric, childbirth experience and childbirth facilities characteristics (n = 269).

Variable	Mean (SD)	n (%)
Parity		
1		87 (32.3)
2–3		124 (46.1)
4 or more		57 (21.3)
Types of facility		
Public childbirth facilities		169 (62.8)
Private childbirth facilities		100 (37.2)
Weeks postpartum		
0–7 weeks		110 (40.9)
8–16 weeks		159 (59.1)
Sex of providers who conduct the deliveries		
Female		215 (79.9)
Male		53 (19.7)
Type of delivery		
Vaginal no episiotomy/tear		149 (55.4)
Vaginal with episiotomy/tear		118 (43.9)
Receiving pain killer during childbirth		
Yes		129 (48)
No		140 (52)
Nature of labor		
Spontaneous		151 (56.1)
Induced		118 (43.9)
Duration of labor (hours)	6.35 (5.529)	
Time of delivery		
Day		163 (60.6)
Night		106 (39.4)

**Table 4 ijerph-19-13180-t004:** Frequency of types of mistreatment experienced by the women during childbirth (n = 269).

Types of Mistreatment	n %
Poor rapport between women and providers	238 (88.5)
Failure to meet professional standards	204 (75.8)
Physical abuse	206 (76.6)
Verbal abuse	66 (24.5)
Health system conditions and constraints	60 (22.3)
Stigma and discrimination	32 (11.9) Total N (269)

**Table 5 ijerph-19-13180-t005:** Simple logistic regression analysis for 13 factors associated with the six types of mistreatment at *p*-value < 0.25.

Variables	Crude OR (95% CI)					
	Physical Abuse	Verbal Abuse	Stigma and Discrimination	Failure to Meet Professional Standard of Care	Poor Rapport between Women and Providers	Lack of Privacy
Age (years)	0.92 (0.87, 0.98) *	0.93 (0.88, 0.99) *	0.89 (0.82, 0.98) *	0.97 (0.91,1.02)	0.96 (0.89, 1.03)	0.96 (0.91, 1.03)
Monthly income (NIS)	1 (1.00, 1.00)	1.00 (1.00, 1.00)	1.00 (1.00, 1.00)	1.00 (1.00,1.00)	1.00 (1.00, 1.00	1.00 (1.00, 1.00)
Parity	0.76 (0.63, 0.93) *	0.90 (0.73, 1.11)	0.69 (0.45, 0.96)	0.92 (0.76, 1.12)	1.11 (0.84, 1.48)	0.85 (0.68, 1.06)
Duration of labor (hours)	1.04 (0.98, 1.11)	1.01 (0.96,1.06)	1.05 (0.99, 1.11)	1.08 (1.01, 1.15) *	1.02 (0.95, 1.10)	1.03 (0.98,1.09)
Education						
Higher than secondary	1	1	1	1	1	1
Secondary or lower	1.46 (0.82, 2.61)	1.42 (0.81, 2.47)	1.01 (0.48, 2.13)	1.70 (0.95, 3.05)	6.10 (2.07, 17.98) *	0.90 (0.51, 1.63)
Residency						
City	1	1	1	1	1	1
Village	1.19 (0.48, 1.48)	0.89 (0.81, 2.47)	2.17 (1.00, 4.69)	1.08 (0.53, 1.63)	1.61 (0.75, 3.46)	0.57 (0.99, 3.17)
Occupation						
Yes	1	1	1	1	1	1
No	0.59 (0.28, 1.25)	0.68 (0.61, 3.50)	0.88 (0.37, 3.44)	0.85 (0.37, 1.96)	2.02 (0.80, 5.01)	1.99 (0.74, 5.35)
Nature of labor						
Spontaneous	1	1	1	1	1	1
Induction	2.60 (1.40, 4.84) *	1.09 (0.62, 1.90)	0.99 (0.47, 2.09)	1.34 (0.76, 2.37)	0.52 (0.25, 1.12)	0.82 (0.46, 1.46)
Time of delivery						
Day time	1	1	1	1	1	1
Night time	1.08(0.82, 3.04)	0.92 (0.52, 1.63)	0.78 (0.36, 1.61)	0.97(0.51, 2.24)	1.42 (0.64, 3.15)	0.79 (0.43, 1.43) *
Receiving painkiller						
Yes	1	1	1	1	1	1
No	0.65 (0.36, 1.15)	1.88 (1.06, 3.32)	1.63 (0.76, 3.47)	1.26 (0.72, 2.20)	5.37 (2.12, 13.57)	2.18 (1.20, 3.98)
Type of delivery						
Vaginal no episiotomy/tear	1	1	1	1	1	1
Vaginal with episiotomy/tear	2.32 (1.27, 4.23) *	1.09 (0.63, 1.90)	3.56 (1.58, 8.03) *	1.45 (0.82, 2.57)	0.30 (0.13, 0.67) *	1.17 (0.66, 2.07)
Type of facility						
Private	1	1	1	1	1	1
Public	1.66 (0.93, 2.94)	3.3 (1.54, 5.84) *	10.39 (2.43, 44.5) *	0.3 (0.19, 0.59)	7.45 (3.08, 8.06) *	2.85 (1.43, 5.69) *
Sex of provider conducting delivery						
Female	1	1	1	1	1	1
Male	1.09 (0.53, 2.22)	0.74 (0.36, 1.54)	1.38 (0.31, 1.71)	0.70 (0.36,1.36)	0.57 (0.25, 1.32)	0.75 (0.35, 1.60)

* significant at *p* < 0.05.

**Table 6 ijerph-19-13180-t006:** Multiple logistic regression analysis of factors associated with each type of mistreatment.

Type of Mistreatment	Significant Variables	Adjusted OR (95% CI)	*p*-Value
Physical abuse	Age	0.91 (0.86, 0.97)	0.003
	Nature of labor		0.001
	Spontaneous	1	
	Induction	3.03 (1.58, 5.81)	
	Type of facility		0.013
	Private	1	
	Public	2.17 (1.18, 4.02)	
Verbal abuse	Age	0.92 (0.86, 0.98)	0.014
	Type of facility		0.001
	Private	1	
	Public	3.25 (1.65 6.41)	
Stigma and discrimination	Type of facility		<0.001
	Private	1	
	Public	16.78 (3.79, 74.26)	
	Type of delivery		<0.001
	Vaginal with no episiotomy/tear	1	
	Vaginal with episiotomy /tear	5.98 (2.52, 14.15)	
	Residency		0.047
	City	1	
	Village	2.33 (1.01, 5.37)	
Failure to meet professional standards of care	Type of facility		<0.001
	Private	1	
	Public	3.31 (1.83, 5.97)	
	Duration of labor (hours)	1.099 (1.02, 1.18)	0.011
Poor rapport between women and providers	Type of facility		0.001
	Private	1	
	Public	4.61 (1.82, 11.71)	
	Education		0.004
	Higher than secondary	1	
	Secondary or lower	5.09 (1.67, 15.57)	
	Receiving painkiller		0.010
	Yes	1	
	No	3.63 (1.36, 9.71)	
Lack of physical privacy and resources	Type of facility		0.007
	Private	1	
	Public	2.75 (1.33, 5.68)	
	Receiving painkiller		0.020
	Yes	1	
	No	2.18 (1.13, 4.22)	
	Parity	0.72 (0.57, 0.93)	0.010

## Data Availability

The data presented in this study are available upon request from the corresponding author.

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
