# Peer review of "Mistreatment of Women during Childbirth and Associated Factors in Northern West Bank, Palestine"

_ijerph, 2022, doi:10.3390/ijerph192013180_

Round 1

Reviewer 1 Report (New Reviewer)

This article deals with an important area of research which has the potential to help change the way women are treated in childbirth and to reduce trauma and abuse in a specific setting. The article presents novel research in the Palestinian West Bank conducted with women who experienced childbirth in the preceeding 16 weeks. The method was an Arabic language questionnaire developed from existing literature and a qualitative study and administered through the public maternal healthcare system. The study found an enormous prevalence of mistreatment in childbirth. It makes recommendations to change this by improving obstetric healthcare.

I was very interested to read this article, which was a strong piece of quantitative research clearly meeting a gap in knowledge about women’s experiences of childbirth in Palestine and attempting to describe the scale of mistreatment. Overall, more could be made of the important findings, the analysis of them, and the conclusions that can be drawn. The scale of mistreatment found in the survey was enormous, and this finding is somewhat underplayed in the abstract and article. I was also shocked by the prevalence of birth without pain relief, which I discuss further below, and which I think should be a finding related to mistreatment rather than a factor. Highlighting these two points could make an even stronger contribution to literature, particularly if the concept of obstetric violence is used or the work is situated alongside obstetric violence and human rights (see below).

It would be useful to have some background information on Palestine / the Occupied Territories early in this article. It is relevant to the findings, because for example residency is a significant variable but residency is very specific to the political situation in relation to Israel. Similarly, limited resources in healthcare such as overcrowding are also connected to the wider political situation. There is a very brief mention of UNWRA in the discussion, but no explanation of why the UN are involved in healthcare in this context. At present, someone reading the article with little knowledge of the conflict in the Middle East would not be able to fully understand the findings. Similarly, a brief description of the healthcare system would be helpful – the findings show that there is a difference between private and public healthcare facilities in terms of outcome and failure to meet professional standards of care, but why people end up in different facilities and how the public system is structured and funded (and how this is affected by the conflict and occupation) is not explained. This is important contextual information. The liminal international status of Palestine is so specific and connected to citizenship / the right of people and states to exist that a discussion of reproduction (amongst other things, the production of new citizens) needs be situated in this context.

The paper was situated in recent literature related to the mistreatment of women in childbirth and the citations were appropriate for this concept, for example Bohren et al 2015. However, the mistreatment concept should reflect or be situated alongside other associated ideas, particularly the concept of obstetric violence, and women’s rights as human rights. In this article, the desirability of eradicating mistreatment of women in healthcare related to birth is framed in terms of improving maternal outcomes and maternal health improvement in general, or in relation to ‘satisfaction’ by patients (eg line 375). This is a limited goal which misses the political and moral elements of ending serious and abusive mistreatment of women at a vulnerable time in their lives. Discussion, for example, of blaming women for not tolerating the pain of episiotomy is focused on women being frustrated and unsatisfied with this – but it clearly shows an abuse of power by care givers, which is much more serious an infringement of autonomy and rights. The article could be much stronger in its condemnation of abuse.

The article does not discuss how the mistreatment of women is potentially situated in political structures in medicine and society, such as patriarchy, medicalisation, class etc. The introduction mentions that structural factors such as age and poverty are associated with mistreatment, but this is taken as a given rather than critically examined. The methods analyse socio-demographic characteristics such as age, education, income etc and the discussion shows that older and more educated women face less mistreatment – but this is not discussed in relation to power relations in the delivery room. The theoretical and social angle could be strengthened by bringing in the concept of obstetric violence which was developed in South America – see Perez D’Gregorio 2010, Sadler et al 2016, Williams et al 2018.. Studies situating obstetric violence in relation to coerced procedures in childbirth (Borges 2017) or in relation to vaginal examination (Shabot 2020) may be useful.

There is also a brief mention of women’s rights in the article (eg citation of Freedman et al 2014, brief mention in the discussion) but this is never developed – Khosla et al 2016, which builds on Bohren, is a useful reference in relation to human rights and mistreatment childbirth. Making more of the rights aspect would fit well with the scale of the mistreatment problem and would make the paper more powerful.

In the methods section, women being approached by midwives in clinic settings is described as ‘random’ selection of participants. It is not possible that such selection would be truly random, since all human interactions contain bias and subjectivity. It would be better to remove this description and just say that midwives approached women, with some comment about the limitations of this method regarding representativeness. Similarly, a comment on the limitations of recruiting within clinics would be useful – the fact that the type of facility is one of the findings which increased vulnerability to mistreatment is significant if women were recruited through these facilities.

Ethics procedures are clear and reported well in the article.

In the results section, lack of pain relief during childbirth was described as an obstetric and childbirth characteristic / variable in this study, rather than a finding in relation to mistreatment. This decision needs clear explanation in the text, or, more usefully, this element moved into the findings. The fact that most women did not receive pain relief is mistreatment in itself. Bohren et al, cited elsewhere in the paper, list refusal to provide pain relief as a form of failure to meet professional standards of care. Placing this characteristic alongside the type of childbirth facility and parity of participants, as this paper does, seems inappropriate as it is likely a result of the decisions of caregivers in the period of childbirth rather than a structural factor such as age or education. Lack of pain relief fits better alongside physical abuse such as painful vaginal examinations. The paper also puts type of facility and education alongside receiving painkillers in relation to poor rapport (line 276) – these are not comparable factors because facility and education levels relate to structural factors, whereas lack of pain relief is a decision made in the moment at an interpersonal level. It is more closely connected to the outcomes described in the paragraph of lack of supportive care, loss of autonomy. The paper would be strengthened if pain relief was moved into findings – set alongside the high levels of prevalence of mistreatment which the study found, the high levels of birth without painkiller make the findings very shocking and significant.

The sentence at line 391 which says ‘women consider’ pain relief to be important in childbirth because it ‘decreases stress’ is problematic – it implies that pain relief is not ‘really’ necessary for childbirth, it just calms women down. It implies women are making a fuss. Such an approach is paternalistic and potentially part of the problem of how women come to be mistreated and unheard in obstetric care. This could easily be rephrased.

The discussion goes on to make recommendations about upgrading childbirth facilities – these points would fit better in the conclusion as they are suggestions derived from the findings, rather than related directly to the findings.

Specific comments

The opening line of the abstract has a typo / missing ‘of’, which detracts from the important statement it makes.

Table 1 was confusing to read – this appeared to be due to the formatting. Items could be lined up more clearly with dividing lines between them.

Author Response

Responses to Reviewer 1 Comments and Suggestions for Authors

This article deals with an important area of research which has the potential to help change the way women are treated in childbirth and to reduce trauma and abuse in a specific setting. The article presents novel research in the Palestinian West Bank conducted with women who experienced childbirth in the proceeding 16 weeks. The method was an Arabic language questionnaire developed from existing literature and a qualitative study and administered through the public maternal healthcare system. The study found an enormous prevalence of mistreatment in childbirth. It makes recommendations to change this by improving obstetric healthcare.

I was very interested to read this article, which was a strong piece of quantitative research clearly meeting a gap in knowledge about women’s experiences of childbirth in Palestine and attempting to describe the scale of mistreatment. Overall, more could be made of the important findings, the analysis of them, and the conclusions that can be drawn. The scale of mistreatment found in the survey was enormous, and this finding is somewhat underplayed in the abstract and article. I was also shocked by the prevalence of birth without pain relief, which I discuss further below, and which I think should be a finding related to mistreatment rather than a factor.

In this study, not receiving Pain relive during childbirth was considered a factor not a form of mistreatment based on the our qualitative finding earlier in this study, so we want to keep it as it is, moreover, the question in the questionnaire related to pain killer (did you receive pain killer or not), so it is a choice for women, may be some women didn’t prefer to take pain killer, so we couldn’t consider this as a form of mistreatment.

I tried to focus more on the importance of pain relieve and rephrased the paragraph as the following

 Given of pain relief to woman is a very important action to help decrease her suffering during childbirth and it is also one of her fundamental rights in case of the presence of pain. In this study, more than half of participants didn’t receive pain killer, this extent the mistreatment of women during childbirth. In fact, deprivation of women from pain killer during childbirth is one form of mistreatment which linked to failure to meet professional standards of care [7]. Other previous studies linked not given painkillers to women during childbirth to poor communication by healthcare providers as they react poorly to labor pain and underestimate the women's sensation of pain. Thus, the women considered healthcare providers as uncooperative and unfriendly. These results are consistent with the findings of studies conducted elsewhere [33,36,37]. Women who did not receive painkillers when needed would experience more pain and difficulty in childbirth, reflecting a negative childbirth experience. A one previous qualitative study conducted among women showed how important the given pain relief to them is necessary element as it decreases their pain and stress during childbirth [38]. Adopting and focusing on the policy of reducing pain relief during childbirth at Palestinian public childbirth facilities is one of important step that decreases women’s burdens during this sensitive period as well as pain relive is one of their rights.

 Highlighting these two points could make an even stronger contribution to literature, particularly if the concept of obstetric violence is used or the work is situated alongside obstetric violence and human rights (see below).

The article is improved and pointed to women’s rights as the following

In abstract:

Abstract: Mistreatment of women during childbirth is a clear breach to women’s rights during childbirth. This study aimed to determine the prevalence and associated factors of mistreatment of women during childbirth in the north of West Bank, Palestine. A cross-sectional study was conducted among 269 women within the first 16 weeks of their last vaginal childbirth to understand the childbirth events by using proportionate stratified random sampling. An Arabic valid questionnaire was used as a study instrument. Simple and multiple logistic regression analyses were conducted to determine the factors associated with each type of mistreatment. The mean age of the women was 26.5 (SD 4.77) years. The overall prevalence of mistreatment was 97.8%. There were six types of mistreatment. Nine factors were significantly associated with the occurrence of one or more types of mistreatment. Delivery at public childbirth facility was associated with all of the six types (aAdjOR: 2.17 – 16.77; p-values < 0.001–0.013). Women who lived in villages (aAdjOR 2.33; p-value = 0.047), had low education (aAdjOR 5.09; p-value = 0.004), underwent induction of labour (aAdjOR 3.03; p-value = 0.001), had a long duration of labour (aAdjOR 1.10; p-value = 0.011), did not receive pain killer (aAdjOR: 2.18 – 3.63; p-values = 0.010–0.020), or had an and having episiotomy or tear (aAdjOR 5.98; p-value < 0.001) were more likely to experience one or more types of mistreatment. With every one hour increase in the duration of labor, they were 1.099 time more likely to experience failure in meeting professional standard of care. Women were less likely to experience mistreatment with increasing age. Women with increasing age (aAdjOR: 0.91 – 0.92; p-values = 0.003–0.014) and parity (aAdjOR 0.72; p-value = 0.010) were less likely to experience mistreatment. Awareness to women’s fundamental rights during childbirth, making the childbirth process as normal as possible, improving the childbirth facilities conditions, policies, practices and working environment may decrease mistreatment occurrence.

In introduction also, I talked about the women’s rights during childbirth and obstetric violence

  In spite of the extent of mistreatment of women during childbirth which donates to a significant breach of women’s fundamental rights [1,2], mistreatment is still insufficiently addressed under international human rights law [2]. In seeking and receiving care before, during and after childbirth, every woman is entitled to several rights. They include the rights to; 1) be free from harm and ill treatment, 2) information, informed consent and refusal, and respect for her choices and preferences, including companionship during childbirth, 3) privacy and confidentiality, 4) treatment with respect and dignity, 5) equality, non-discrimination, and equitable care, 6) healthcare and high achievable level of health, and 7) liberty, autonomy, self-determination and non-coercion [3].

Unfortunately, the presence of mistreatment of women during childbirth has been confirmed globally [4- 14]. Evidence showed that a third of women are exposed to verbal or physical abuse, during childbirth [4].  

  Mistreatment of women during childbirth is a complicated phenomenon and its definitions are heterogeneous across the literature due to the variation in the types and terms used. The most commonly employed term is “disrespect and abuse,” which was introduced in landscape analysis [5]. It was defined as “interactions or facility conditions that local consensus deem to be humiliating or undignified, and those interactions or conditions that are experienced as or intended to be humiliating or undignified” [6].

The term “mistreatment” can be used instead of “disrespect and abuse” as it is more comprehensive because it covers a wide scope of categories and emphases on different sources of mistreatment [7]. ‘’Obstetric violence’’ is another term that is also used in Latin America and the Caribbean to describe abuse or mistreatment during childbirth that women face from health care providers [8]. The three terms; ’obstetric violence’’, ‘’disrespect and abuse’’ and mistreatment are sharing the concepts (medicalization of childbirth process, gender inequality) alongside with violence against women [9]. The mentioned three concepts are used to explain the extent of mistreatment.

The use of seven evidence-based types of mistreatment was recommended in a systematic review that encompassed physical abuse, sexual abuse, verbal abuse, failure to meet professional standards of care, stigma and discrimination, poor rapport between women and providers, and healthcare system conditions and constraints [7]. These types are recommended in developing new tools for better measuring and avoid underestimation of the prevalence of mistreatment [7].

The global mistreatment prevalence ranged from 11 to 98%, as reported in previous studies (e.g., 98% in Nigeria [10], 78% in Kenya [11], 97% in Pakistan [12], 28.8% in India [13], 49.4% in Latin America [14], 18.3% in Brazil [15], and 11% in Mexico [16]). Mistreatment prevalence may vary due to cultural differences in the participants’ setting of origin, in addition to childbirth conditions and restraints. Moreover, the manifestations and detailed assessment of the types of mistreatment presented, as well as the timing and place of data collection, influenced the reported prevalence [17].

Certain factors were found to be associated with mistreatment of women during childbirth. Those factors included young age mother [4,15,18,19] or being single mother [20], being uneducated [4,10], low socioeconomic status [10,121,22] and those living in poor residency [23]. On the other hand, obstetric characteristics, such as the presence of complications during childbirth, sex of provider [13] and women’s parity were also reported as factors associated with the occurrence of mistreatment [7,11,13]. An additional factor was giving birth at public childbirth facility [7, 13,15,24].

It would be useful to have some background information on Palestine / the Occupied Territories early in this article. It is relevant to the findings, because for example residency is a significant variable but residency is very specific to the political situation in relation to Israel. Similarly, limited resources in healthcare such as overcrowding are also connected to the wider political situation. There is a very brief mention of UNWRA in the discussion, but no explanation of why the UN are involved in healthcare in this context. At present, someone reading the article with little knowledge of the conflict in the Middle East would not be able to fully understand the findings. Similarly, a brief description of the healthcare system would be helpful – the findings show that there is a difference between private and public healthcare facilities in terms of outcome and failure to meet professional standards of care, but why people end up in different facilities and how the public system is structured and funded (and how this is affected by the conflict and occupation) is not explained. This is important contextual information. The liminal international status of Palestine is so specific and connected to citizenship / the right of people and states to exist that a discussion of reproduction (amongst other things, the production of new citizens) needs be situated in this context.

Palestinian context is added to the manuscript under the introduction covering most of the above-mentioned points as the following

Palestinian context

Palestine is country which is located in Asia between the Mediterranean Sea and the Jordan River. Historically, it is known that a part of Palestinian land has been occupied by Israel since 1948. Since 1948, the political situation of Palestine has been unstable and the Palestinian people have been striving to get their fundamental rights as human beings, get back their land and live in Pease and dignity.

In 1967 Israel occupied the rest of Palestine which includes West Bank and Gaza that is known now the Occupied Palestinian Territory (OPT). The Palestinian Authority (PA) was found in 1994 which was given some control over the OPT land and so it was responsible for governing the health care system. The United Nations Relief and Works Agency (UNRWA) has been found by the United Nations since 1949 as a consequence of Israeli occupation for the purpose of providing direct relief and works programmes for Palestinian refugees.  Since that time the UNRWA has shared in the Palestinian health care system.

The political situation in Palestine has been negatively affected the health care system especially the public childbirth facilities.  So, this system became very poor with a low quality [25], chronic deficiency of medical supplies and vital medical disposable which affects certain areas and treatment pathways [26].

In fact, the public childbirth facilities consider the convenient choice for the women with low and middle-income because of the availability of the health insurance coverage while the women with high-income utilize the private childbirth facilities. The financial status of the family plays an important role on women choices of type of facility. 

 Unfortunately, the Palestinian community has special considerations due to political situation, thus Palestinians always suffer from violation of human rights and experience feelings of fears, worries, and threats in their daily life [27]. With regards the Palestinian women who often suffer from this Israeli occupation, they have extra difficulty in their lives because when experiencing labour pain, women usually have unsafe and insecure way to reach childbirth facilities due to the presence of roadblocks [25]. Previous Literature also showed that the period from 2000 to 2006, 69 Palestinian women delivered on the check points in West Bank under unsafe and dehumanizing conditions [28].

Israeli occupation and the political instability in Palestine are one of the main hiding factors of all the abusive mistreatment of Palestinian women during their lives especially during childbirth. Mistreatment during childbirth is a part of violence against women that women may be exposed in their lives and it also considers a major violation of women’s fundamental rights. Actually, Palestinian women had unique experiences, facing multiple types of abusive mistreatment at the same time. Investigating mistreatment among these women is very necessary as it will be the first step for raising awareness of their fundamental rights that they deprived from because of the political conflict towards ending serious and abusive mistreatment of Palestinian women at a vulnerable time in their lives. The aim of this study is to measure the prevalence of the mistreatment of women during childbirth and its associated factors in the northern West Bank, Palestine.

The paper was situated in recent literature related to the mistreatment of women in childbirth and the citations were appropriate for this concept, for example Bohren et al 2015. However, the mistreatment concept should reflect or be situated alongside other associated ideas, particularly the concept of obstetric violence, and women’s rights as human rights.

I tried to point to the concept of the women’s rights in the abstract and the women’s rights and obstetric violence in the introduction and manuscript and in discussion

In this article, the desirability of eradicating mistreatment of women in healthcare related to birth is framed in terms of improving maternal outcomes and maternal health improvement in general, or in relation to ‘satisfaction’ by patients (eg line 375). This is a limited goal which misses the political and moral elements of ending serious and abusive mistreatment of women at a vulnerable time in their lives.

The satisfaction’ by patients (in line 375) rephrased and the goal is improved as the following under Palestinian context

Investigating mistreatment among these women is very necessary as it will be the first step for raising awareness of their fundamental rights that they deprived from because of the political conflict towards ending serious and abusive mistreatment of Palestinian women at a vulnerable time in their lives. The aim of this study is to measure the prevalence of the mistreatment of women during childbirth and its associated factors in the northern West Bank, Palestine.

Discussion, for example, of blaming women for not tolerating the pain of episiotomy is focused on women being frustrated and unsatisfied with this – but it clearly shows an abuse of power by care givers, which is much more serious an infringement of autonomy and rights. The article could be much stronger in its condemnation of abuse.

Yes, the paragraph is rephrased as the following

The longer the duration of labor, the more likely women are to experience negligence of care during childbirth, lack of confidentiality, and non-consented care. This is also agreed with findings of a Swedish study showing that prolonged labor is a contributing factor to negative childbirth experience [40].

Furthermore, having vaginal delivery with the experience of an episiotomy or a tear is also another an associated factor of mistreatment. This may be explained that women loss their control over their bodies and they also feel that their rights were disregarded like privacy and autonomy during episiotomy procedure adding to that the pain women experience during and after this procedure. Like these complains was also reported by a recent Chinese qualitative study. Women who experienced an episiotomy reported that they were being criticized by healthcare providers for not tolerating the pain during the procedure [37]. Other studies discovered that women who experienced perineal trauma were neglected by healthcare providers during the procedure [41,42]. In fact, the health care providers sometimes use their power they taught from health care system when dealing with women during childbirth to control them. Actually, taking coercive procedures to women by the health care providers clearly shows an abuse of power by care givers, which is much more serious violation of women autonomy and rights [2,6]. 

The article does not discuss how the mistreatment of women is potentially situated in political structures in medicine and society, such as patriarchy, medicalisation, class etc. The introduction mentions that structural factors such as age and poverty are associated with mistreatment, but this is taken as a given rather than critically examined.

The discussion revised to cove these points political situation, medicalization, the structure of the health care systems

Multiple hidden factors play an important role in the extent of mistreatment of women in Palestine and the violation of women's rights during childbirth. These factors include the political situation in Palestine, the structure of the Palestinian health care system and its futile policies, medicalization to normal childbirth process and society culture, belief, gender inequalities and male dominance.

 The political situation in Palestine and its negative effects on the health care system resulted in poor public childbirth facilities environments. The problems related to poor childbirth facilities environments are common and reported not only in the Palestinian healthcare system but also in other countries, especially developing ones [5,7, 10,13, 21,24,32, 33].

In this study, delivery at public childbirth facility was found to be associated with all the six types of mistreatment; physical abuse, verbal abuse, stigma and discrimination, poor rapport between women and providers, failure to meet professional standards of care, and health system conditions and constraints.  This means, several of a woman’s rights are violated when she gets birth at public childbirth facility as she is exposed to the six types of mistreatment. Regardless the numerous health services provided by public childbirth facilities, they are still facing long-standing insufficiency of medical supplies and vital medical disposables, which interferes with certain areas of care and treatment pathways [34]. In fact, there is no isolation between the poor conditions of the health care system in West Bank from the political situation in Palestine because Israeli occupation is controlling the resources and putting restrictions on the improvement and the outside funding. The Public childbirth facilities are also lack of good quality services due to the high number of deliveries and crowded labor rooms [28,35]. This may result in overload work for the healthcare providers, which affects their performance and is reflected in the quality of care provided to women. Additionally, a recent policy prevents the presence of a birth companion during delivery; thus, the women have to spend all the time alone in the labor room and feel neglected. Previous studies support the present study’s finding that public childbirth facilities were associated with a high prevalence of mistreatment during childbirth [7,13, 15, 24].

Furthermore, the structure of the Palestinian health care system and its futile policies are other contributing factors to mistreatment of women during childbirth.  For instance, the policy that prevents the presence of a birth companion which has been existed in public childbirth facilities and women have complained from for a long time. Denial of companionship during childbirth has been shown to cause women to feel disempowered and lonely [7]. African women described the disallowing of a childbirth companion as a ‘crime against humanity’ [10]. In addition to that, the presence of ineffective monitoring systems and the lack of accountability mechanisms, and the non-adherence of healthcare providers to evidence-based practices during the provision of care may worsen the situation [5, 7,12,14, 29].

Given of pain relief to woman is a very important action to help decrease her suffering during childbirth and it is also one of her fundamental rights in case of the presence of pain. In this study, more than half of participants didn’t receive pain killer, this extent the mistreatment of women during childbirth. In fact, deprivation of women from pain killer during childbirth is one form of mistreatment which linked to failure to meet professional standards of care [7]. Other previous studies linked not given painkillers to women during childbirth to poor communication by healthcare providers as they react poorly to labor pain and underestimate the women's sensation of pain. Thus, the women considered healthcare providers as uncooperative and unfriendly. These results are consistent with the findings of studies conducted elsewhere [33,36,37]. Women who did not receive painkillers when needed would experience more pain and difficulty in childbirth, reflecting a negative childbirth experience. A one previous qualitative study conducted among women showed how important the given pain relief to them is necessary element as it decreases their pain and stress during childbirth [38]. Adopting and focusing on the policy of reducing pain relief during childbirth at Palestinian public childbirth facilities is one of important step that decreases women’s burdens during this sensitive period as well as pain relive is one of their rights.

Medicalization to normal childbirth process is highly contributed to the extent of mistreatment as it stands side by side with the concept of obstetric violence because it makes women more vulnerable to painful, harmful and difficult procedures [8], such as induction of labour and episiotomy, frequent vaginal examination [32,39] and long labour duration [39,40]. Medicalization of childbirth process involves denying to women's right to autonomy and consent [2,6]. This is also corresponding with the result of this study, the nature of labor, especially induced labor, is significantly associated with physical abuse. This is because women who undergo induction of labor are frequently forced to undergo various difficult interventions and procedures that have the nature of mistreatment These situations were also mentioned in previous related studies in Jordan [33,36].

The longer the duration of labor, the more likely women are to experience negligence of care during childbirth, lack of confidentiality, and non-consented care. This is also agreed with findings of a Swedish study showing that prolonged labor is a contributing factor to negative childbirth experience [40].

Furthermore, having vaginal delivery with the experience of an episiotomy or a tear is also another an associated factor of mistreatment. This may be explained that women loss their control over their bodies and they also feel that their rights were disregarded like privacy and autonomy during episiotomy procedure adding to that the pain women experience during and after this procedure. Like these complains was also reported by a recent Chinese qualitative study. Women who experienced an episiotomy reported that they were being criticized by healthcare providers for not tolerating the pain during the procedure [37]. Other studies discovered that women who experienced perineal trauma were neglected by healthcare providers during the procedure [41,42]. In fact, the health care providers sometimes use their power they taught from health care system when dealing with women during childbirth to control them. Actually, taking coercive procedures to women by the health care providers clearly shows an abuse of power by care givers, which is much more serious violation of women autonomy and rights [2,6]. 

The extent of mistreatment of women during childbirth is also affected by society culture, belief, gender inequality and male dominance which is very high among the Palestinian community [43].  Actually, the predominant patriarchal culture in Palestinian society is the base for the gender inequality which limits the women from practicing their rights during childbirth that is mainly decided on behalf of them. In this study, age of women, education and parity were found as protective factors of mistreatment. Older women were less likely to experience physical and verbal abuses. The fact that the normalization of verbal and physical abuse among older women makes them more adaptive and less sensitive to tough events during childbirth [5,7]. Some women even considered abuse as a way of accelerating their delivery (5,10,11]. In spite of excluding women younger than 18 years from this study, the results nonetheless showed that the younger the woman, the more exposed she is to mistreatment.

Additionally, with an increase in parity, the women were more habituated to the childbirth process and frequent procedures; thus, they had become less sensitized to the lack of privacy and resources. Their expectations of the care that they would receive during childbirth were lowered. The women who were experienced in childbirth had become accustomed to mistreatment and consider it normal [5]. Moreover, these women felt confidence, had positive attitudes, and enjoyed higher satisfaction because they had the chance to participate in the decision making for their care [44,45]. Actually, the root of normalization of abuse among women during childbirth is derived from the society culture and gender inequality which make women accustomed to be abused during their life time. 

Education is another factor associated with mistreatment, as reported by previous studies [7,23,46]. Women with a low level of education is associated with poor rapport with providers because the providers thought that women might not comprehend the instructions given to them. So, the healthcare provider intentionally displayed abusive behaviour to control the women during childbirth [46]. Accordingly, the invisible causes of mistreatment that women exposed to during childbirth because of their young ages, their low education and lack of childbirth experiences, are due to the predominant socialization of men and women into naturalised, forms of violence and power dynamics between groups [8]. This form of mistreatment is also parallel to violence against women, so the healthcare providers who hold the power in labour room unintentionally abuse women through their authority and other times healthcare providers decide on behalf of the women during childbirth process.

Residential area was also a predictor of mistreatment [23]. Residency of the Palestinian women is naturally a mistreating factor for them because of the Israeli occupation and political instability; the presence of roadblocks and check points that women have to pass so that they can reach the childbirth facilities are significant obstacles for them. Previously, some childbirth took place at check points with dehumanized and unsafe environment [25,28]. Such practices are incongruent to human rights, as women must have the right to access safe and respectful care. Women from rural areas also are affected from lack to access good medical resources which resulted from chronic shortages in Palestinian health care system [26], because the health care system is related - in a way or another - with the political situation in Palestine. 

The methods analyse socio-demographic characteristics such as age, education, income etc and the discussion shows that older and more educated women face less mistreatment – but this is not discussed in relation to power relations in the delivery room.

The theoretical and social angle could be strengthened by bringing in the concept of obstetric violence which was developed in South America – see Perez D’Gregorio 2010, Sadler et al 2016, Williams et al 2018. Studies situating obstetric violence in relation to coerced procedures in childbirth (Borges 2017) or in relation to vaginal examination (Shabot 2020) may be useful.

 I tried to discuss the education, age, parity in relation to power relations in the delivery room and bringing in the concept of obstetric violence using some of the above references

In this study, age of women, education and parity were found as protective factors of mistreatment. Older women were less likely to experience physical and verbal abuses. The fact that the normalization of verbal and physical abuse among older women makes them more adaptive and less sensitive to tough events during childbirth [5,7]. Some women even considered abuse as a way of accelerating their delivery (5,10,11]. In spite of excluding women younger than 18 years from this study, the results nonetheless showed that the younger the woman, the more exposed she is to mistreatment. Actually, women with younger ages, low education and lack of childbirth experiences more prone to mistreatment because of the inside concept from the healthcare providers that these women could be easily controlled referring to their characteristics that mentioned above.

Additionally, with an increase in parity, the women were more habituated to the childbirth process and frequent procedures; thus, they had become less sensitized to the lack of privacy and resources. Their expectations of the care that they would receive during childbirth were lowered. The women who were experienced in childbirth had become accustomed to mistreatment and consider it normal [5]. Moreover, these women felt confidence, had positive attitudes, and enjoyed higher satisfaction because they had the chance to participate in the decision making for their care [44,45]. Actually, the root of normalization of abuse among women during childbirth is derived from the society culture and gender inequality which make women accustomed to be abused during their life time. 

Education is another factor associated with mistreatment, as reported by previous studies [7,23,46]. Women with a low level of education is associated with poor rapport with providers because the providers thought that women might not comprehend the instructions given to them. So, the healthcare provider intentionally displayed abusive behaviour to control the women during childbirth [46]. Accordingly, the invisible causes of mistreatment that women exposed to during childbirth because of their young ages, their low education and lack of childbirth experiences, are due to the predominant socialization of men and women into naturalised, forms of violence and power dynamics between groups [8]. This form of mistreatment is also parallel to violence against women, so the healthcare providers who hold the power in labour room unintentionally abuse women through their authority and other times healthcare providers decide on behalf of the women during childbirth process.

There is also a brief mention of women’s rights in the article (eg citation of Freedman et al 2014, brief mention in the discussion) but this is never developed – Khosla et al 2016, which builds on Bohren, is a useful reference in relation to human rights and mistreatment childbirth. Making more of the rights aspect would fit well with the scale of the mistreatment problem and would make the paper more powerful.

I added the women rights in the introduction and tried to discuss the finding in relation to women’s rights

In spite of the extent of mistreatment of women during childbirth which donates to a significant breach of women’s fundamental rights [1,2], mistreatment is still insufficiently addressed under international human rights law [2]. In seeking and receiving care before, during and after childbirth, every woman is entitled to several rights. They include the rights to; 1) be free from harm and ill treatment, 2) information, informed consent and refusal, and respect for her choices and preferences, including companionship during childbirth, 3) privacy and confidentiality, 4) treatment with respect and dignity, 5) equality, non-discrimination, and equitable care, 6) healthcare and high achievable level of health, and 7) liberty, autonomy, self-determination and non-coercion [3].

In the methods section, women being approached by midwives in clinic settings is described as ‘random’ selection of participants. It is not possible that such selection would be truly random, since all human interactions contain bias and subjectivity. It would be better to remove this description and just say that midwives approached women, with some comment about the limitations of this method regarding representativeness. Similarly, a comment on the limitations of recruiting within clinics would be useful – the fact that the type of facility is one of the findings which increased vulnerability to mistreatment is significant if women were recruited through these facilities.

The descriptions that midwives randomly approached …. Has been removed and, a comment on the limitations of recruiting within clinics has been added and reported as the following

The Palestinian Ministry of Health clinics are considered the main centers for children immunization and these centers can be utilized by women coming from both public and private childbirth facilities.

Before the study was conducted, the data collectors were trained in the correct method of collecting data, in how to understand the questionnaire, and in the proper way of interacting with the participants. The data collectors were also provided with instructions about the objectives of the research, the gathering of sensitive data, and applying ethical principles during data collection. This training was conducted to ensure the standardization of the data collection procedures. The women were approached by the data collectors at the maternal and child health clinics during their visits to vaccinate their newborns. Actually, the interactions between the midwives and the women may include some bias and subjectivity as well as recruiting the women within the Palestinian Ministry of Health clinics may also affect the representativeness of the sample.   

 In the results section, lack of pain relief during childbirth was described as an obstetric and childbirth characteristic / variable in this study, rather than a finding in relation to mistreatment. This decision needs clear explanation in the text, or, more usefully, this element moved into the findings. The fact that most women did not receive pain relief is mistreatment in itself. Bohren et al, cited elsewhere in the paper, list refusal to provide pain relief as a form of failure to meet professional standards of care. Placing this characteristic alongside the type of childbirth facility and parity of participants, as this paper does, seems inappropriate as it is likely a result of the decisions of caregivers in the period of childbirth rather than a structural factor such as age or education. Lack of pain relief fits better alongside physical abuse such as painful vaginal examinations.

The paper also puts type of facility and education alongside receiving painkillers in relation to poor rapport (line 276) – these are not comparable factors because facility and education levels relate to structural factors, whereas lack of pain relief is a decision made in the moment at an interpersonal level. It is more closely connected to the outcomes described in the paragraph of lack of supportive care, loss of autonomy. The paper would be strengthened if pain relief was moved into findings – set alongside the high levels of prevalence of mistreatment which the study found, the high levels of birth without painkiller make the findings very shocking and significant.

During the analysis, I have conducted sex model of logistic regression (all the factors with each of the sex type of mistreatment). The results of logistic regression showed that type of facility, education and not receiving painkillers were associated with poor rapport between women and providers. In discussion I tried to discuss these issues.

In this study, not receiving Pain relive during childbirth was considered a factor not a form of mistreatment based on our qualitative finding earlier in this study, so we want to keep it as it is. Moreover, the question in the questionnaire related to pain killer (did you receive pain killer or not), so it is a choice for women, may be some women didn’t prefer to take pain killer, so we couldn’t consider this as a form of mistreatment.

The sentence at line 391 which says ‘women consider’ pain relief to be important in childbirth because it ‘decreases stress’ is problematic – it implies that pain relief is not ‘really’ necessary for childbirth, it just calms women down. It implies women are making a fuss. Such an approach is paternalistic and potentially part of the problem of how women come to be mistreated and unheard in obstetric care. This could easily be rephrased.

The sentence is rephrased as the following

. A one previous qualitative study conducted among women showed how important the given pain relief to them is necessary element as it decreases their pain and stress during childbirth [38].

 The discussion goes on to make recommendations about upgrading childbirth facilities – these points would fit better in the conclusion as they are suggestions derived from the findings, rather than related directly to the findings.

The recommendations removed from the discussions, corrected and added under the conclusion

  1. Conclusion

The results of this study, one of the first to consider in West Bank, showed that a high percentage of women experienced mistreatment during childbirth. The most common type was poor rapport between women and providers, followed by physical abuse and failure to meet professional standard of care. Age, nature of labor, type of facility, type of delivery, residency, duration of labor, education, receiving painkillers and parity were the factors significantly associated with mistreatment during childbirth. The mistreatment of women during childbirth is caused by multiple factors that negatively affect women’s childbirth experiences. To address the mistreatment revealed by this research, multiple initiatives should be undertaken, such as consideration of the results by stakeholders in the improvement of the environment surrounding childbirth, as well as a massive investment of effort in addressing the factors that lead to the environment of mistreatment, such as staff constraints and poor working conditions. Decision makers should intensify their focus on upgrading childbirth facilities, especially public ones, by improving the childbirth environment in general and the healthcare providers’ working conditions specifically. Related measures may include increasing the number of healthcare providers and motivating them to be more productive and have a more positive attitude toward their work. Additionally, it is vital to improve the conditions surrounding childbirth practices by decreasing unnecessary interventions and striving for a more spontaneous childbirth process. Healthcare providers should provide adequate analgesia during and after the episiotomy procedure to reduce women's pain and suffering. They must not only acknowledge the delivering women's rights but also advocate those rights.

In addition, policies should be modified in favor of considering women’s preferences during childbirth, such as allowing the presence of a companion and keeping childbirth practices free of unnecessary interventions and making the process of childbirth normal as much as possible. Administrators and providers should stress on certain vital principles during childbirth, such as respectful care, pain management, communication skills, ethical principles, and women’s rights. A greater concentration is needed on the establishment of systems of monitoring and ensuring effective accountability at childbirth facilities.

Specific comments

 The opening line of the abstract has a typo / missing ‘of’, which detracts from the important statement it makes.

 Yes, there are missing word. The opening line changed as the following

Abstract: Mistreatment of women during childbirth is a clear breach to women’s rights during childbirth.

Table 1 was confusing to read – this appeared to be due to the formatting. Items could be lined up more clearly with dividing lines between them.

The table has been corrected in the manuscript

Reviewer 2 Report (New Reviewer)

Dear Authors, I thank you for the opportunity to read this manuscript, as I believe it addresses very important issue, that has arisen as a public health issue in multiple countries, but that has not been very well examined in many regions and is still not frequently an option for the research topic.

Please find the minor comments for the authors to correct:

Introduction

Line 39: instead ,,one third of women exposed’’, should be ,,one third of women are exposed’’

Lines 68-69: ,,Since the eradication of mistreatment…’’, please rephrase the sentence, as this was not shown in the introduction so far. Maybe start with ,,As the previous studies have shown that..’’

Methods

Study instrument

Line 99: add that it was developed through the qualitative study

Data collection

Please add the response rate. 

Author Response

Comments and Suggestions for Authors

Introduction

Line 39: instead,, one third of women exposed’’, should be ,,one third of women are exposed’’

It has been corrected as requested

Unfortunately, the presence of mistreatment of women during childbirth has been confirmed globally. Evidence showed that a third of women are exposed to verbal or physical abuse, during childbirth [4]. 

Lines 68-69:,, Since the eradication of mistreatment…’’, please rephrase the sentence, as this was not shown in the introduction so far. Maybe start with,, As the previous studies have shown that..’’

This sentence was removed from the introduction because I have done some changes requested by another reviewer.

Methods

Study instrument

Line 99: add that it was developed through the qualitative study

It has been added as requested.

 A pretested self-administered mistreatment during childbirth questionnaire developed in the Arabic language and validated among women during their first 16 weeks postpartum in West Bank, Palestine, it was developed through the qualitative study and used for the purpose of measuring the experience of mistreatment of women during childbirth,

Data collection

 Please add the response rate. 

It has been added as requested.

The number of questionnaires distributed in each governorate was based on the calculated ratio previously explained. with a response rate of 100%.

This manuscript is a resubmission of an earlier submission. The following is a list of the peer review reports and author responses from that submission.

Round 1

Reviewer 1 Report

Thank you for giving me the opportunity to review this manuscript. It is indeed very interesting and addresses a very trendy topic in European countries. I understand this has not been evaluated in the past in Palestine, so it is of relevance to publish these results. I would like to share with the authors some minor comments about the background, methods and discussion, and some other major issues on the results. Thank you.

Background:

-          I would suggest re-organizing the background and explaining what the authors mean by ‘mistreatment’ and ‘types of mistreatment’ since the very beginning. I felt a bit lost inline 32, for example, where they talk about ‘various types [of mistreatment]’ but do not provide details on those after two paragraphs.

-          Line 45: Please, consider re-phrasing to “The use of 7 evidence-based…”. The sentence can be misinterpreted if you say that there are 7 types of mistreatment that are recommended, instead of the use of that terminology.

-          Lines 53-55 suggest that women’s attributes are causes or aggravating factors of mistreatment, please re-phrase to clearly indicate those are just factors associated with increased prevalence of mistreatment.

Methods:

-          Lines 81-86: Please justify why you decided to use a proportionate stratified random sampling instead of giving a different weight to each facility/community based on their number of inhabitants. If one of the facilities is used to attend a much higher number of individuals, that should have a higher number of study participants to obtain a final prevalence figure that represents the whole population. This should also be discussed further in the discussion.

-          Section 2.2 Study Instrument: validated among which type of women? Pregnant women? Postpartum women?

-          Please, publish the questionnaire as a supplementary online file.

-          Line 109 seems to say that questionnaires where self-administered by women. Please correct that sentence, right before you say those where administered by data collectors (midwives).

-          Line 111: please, provide in the methods section the specific enrollment dates for each clinic/facility since not all of participants where enrolled during the same months (enrollment in some facilities happened during end of 2020, while other occurred in mid 2021).

-          Was data collected in paper format and then entered by one individual? Please make this clear, and consider mentioning the potential data entry errors made since there was no double data entry as it would have been optimal.

-          Line 144: six multiple logistic regression analysis, I think the word ‘multiple’ is missing.

Results:

-          Line 148: Why were participants only married women? If it was just by chance, please include ‘marital status’ in Table1. If this was by design, please include ‘marital status among the inclusion criteria.

-          Table 2: correct ‘type of provider’ to ‘sex/gender of provider’.

-          Table 3: please add total N.

-          Regressions: I recommend checking ‘Facility where women delivered’ in the multivariable models, not only type of facility. I would adjust the analysis or use random effects/clustering of SE to account for differences across facilities.

-          Table 4: what the * means?

-          Table 5: please, publish complete results of the regressions in an online appendix, not only significant OR and p-values.

Discussion:

-          Line 253: was this due to COVID-19 restrictions? The impact of COVID-19 pandemic is not discussed in the paper and I think it is very important since the study was performed during the first year of the pandemic, and that could have biased the results.

-          Lines 261-267: discuss the exclusion of adolescents in relation to the association age-mistreatment.

-          Although it is briefly mentioned in the last paragraph of the discussion, it is necessary to discuss more about the impact of the exclusion criteria on the results, since excluded women were those at a higher risk of mistreatment following the information provided in the background.

Author Response

Please see the attachment in the box

Reviewer 2 Report

This is a study that assesses the prevalence and associated factors of mistreatment during childbirth in West Bank, Palestine. The overall prevalence of mistreatment was about 98%, and the associated factors were poor rapport between women and providers followed by physical abuse and failure to meet professional standard of care. There are several comments and concerns below for authors' consideration in order to further improve the quality of the paper.

Title: This should be revised to reflect the study population being investigated; is it pregnant women or the healthcare providers?

Abstract: In line 13, it is stated that pregnant women were studied within the first 16 weeks postpartum (after delivery). However, in lines 69 and 90, intrapartum (during child birth) data were stated to have been collected for duration of labor, mode of labor, and receipt of analgesic during childbirth. In the title, abstract, and introduction, the authors should state clearly the actual period around childbirth for which data were collected (prepartum, intrapartum and postpartum???). Use the abbreviation "aOR" instead of "AdjOR". Also include the confidence intervals or p-values for the reported adjusted ORs in the abstract. In line 24, revise the phrase "...improving the childbirth facilities conditions..."

Introduction: Some errors in English and grammar are highlighted for revision in the attached file. 

Methods: In line 76, the statement "The inclusion criteria were women during the first 16 weeks of their vaginal deliveries..." is not consistent with the statement in the abstract. Revise this. Also, what are the reasons for excluding women with multiple pregnancy, and stillbirth from the study? In line 107 revise the phrase " They were explained about the study and reassured.." In line 113, also revise "This study obtained ethical approval..."

Statistical analysis: It is important to include what type of questions in the survey that were asked for some variables like "failure to meet professional standard of care", and "health care system conditions and constraints". In line 136, how were clinically important variables determined?

Results: In lines 149-150, the numbers for education do not tally with those reported in table 1. Revise this. The variable "type of care provider who conduct deliveries" should be properly labeled as "Sex of care provider conducting deliveries". In table 2, no data are recorded for the variable "Time of delivery". Explain the meaning of the terms "loss of autonomy during childbirth" and "Failure to meet professional standard of care". How does the untrained patient know if care is standard or not? What questions were asked for these variables? In line 184, explain "clinical significance". In line 187, revise "...received of pain killer..."

It is difficult to understand how the variables that were not significantly associated with mistreatment in the simple logistic regression suddenly had significant associations after adjustments in the multiple logistic regression.

Conclusion: This section needs to be revised. The authors have only summarized the study findings here, but have failed to include some of the suggested solutions to the problem, and public health implications.

There are several errors in English and grammar that need to be thoroughly revised, if possible, with the help of an English editor. Some of the errors are highlighted for revision in the attached file.

Best of luck!!!

Author Response

Please see the attachment in the box

Round 2

Reviewer 1 Report

Thank you for improving the manuscript and responding to all questions. I just have a few minor follow-up comments for your consideration:

- Table 3: I would include the N in a row at the bottom of the table instead of in a new column.

- Regressions: I understand that 'type of facility' refers to the facilities where women delivered. However, my point was to adjust the analysis by the specific facility, not the type. There could be some clustering of SE, for example.

- I cannot find the complete results of the regressions as a supplemental material. Please publish those if possible.

- Questionnaires: for a broader understanding and use of the questionnaires you used, I suggest uploading English translations.

Thank you!

Reviewer 2 Report

The authors have adequately addressed most of my concerns from the initial review, although I am still uncomfortable with the results from the logistic regression analyses despite the author's explanation.
